# Lasting Peripheral and Central Effects of Botulinum Toxin Type A on Experimental Muscle Hypertonia in Rats

**DOI:** 10.3390/ijms231911626

**Published:** 2022-10-01

**Authors:** Petra Šoštarić, Barbara Vukić, Lea Tomašić, Ivica Matak

**Affiliations:** 1Department of Pharmacology, University of Zagreb School of Medicine, Šalata 11, 10000 Zagreb, Croatia; 2University Psychiatric Hospital Vrapče, University of Zagreb School of Medicine, Bolnička Cesta, 10090 Zagreb, Croatia

**Keywords:** botulinum toxin type A, antispastic activity, duration of action, central effect

## Abstract

Recent animal experiments suggested that centrally transported botulinum toxin type A (BoNT-A) might reduce an abnormal muscle tone, though with an unknown contribution to the dominant peripheral muscular effect observed clinically. Herein, we examined if late BoNT-A antispastic actions persist due to possible central toxin actions in rats. The early effect of intramuscular (i.m.) BoNT-A (5, 2 and 1 U/kg) on a reversible tetanus toxin (TeNT)-induced calf muscle spasm was examined 7 d post-TeNT and later during recovery from flaccid paralysis (TeNT reinjected on day 49 post-BoNT-A). Lumbar intrathecal (i.t.) BoNT-A–neutralizing antiserum was used to discriminate the transcytosis-dependent central toxin action of 5 U/kg BoNT-A. BoNT-A-truncated synaptosomal-associated protein 25 immunoreactivity was examined in the muscles and spinal cord at day 71 post-BoNT-A. All doses (5, 2 and 1 U/kg) induced similar antispastic actions in the early period (days 1–14) post-BoNT-A. After repeated TeNT, only the higher two doses prevented the muscle spasm and associated locomotor deficit. Central trans-synaptic activity contributed to the late antispastic effect of 5 U/kg BoNT-A. Ongoing BoNT-A enzymatic activity was present in both injected muscle and the spinal cord. These observations suggest that the treatment duration in sustained or intermittent muscular hyperactivity might be maintained by higher doses and combined peripheral and central BoNT-A action.

## 1. Introduction

Botulinum neurotoxin type A (BoNT-A) and other immunogenically distinct serotypes (BoNT-B–BoNT-G) are potent clostridial neurotoxins that cause botulism, a potentially fatal neuroparalytic illness characterized by the long-term synaptic silencing of peripheral muscular and autonomic nerve terminals [1,2]. Their extreme potency (estimated LD_50_ of BoNT-A is around 1 ng/kg [2,3]) due to selective nerve terminal uptake and inhibition of the neurotransmitter release machinery within the presynaptic active zones [4,5]. The BoNT-A serotype enzymatically cleaves off only a small C-terminal part (amino-acids 197–206) of synaptosomal-associated protein 25 (SNAP-25) involved in the Ca^2+^-triggered synaptic vesicle exocytosis [5]. Used as a standardized low-dose pharmaceutical preparation, locally injected BoNT-A (A1 subtype BoNT synthesized by the *Clostridium botulinum* strain Hall) has become the first-choice treatment of several neurological disorders characterized by sustained or intermittent focal muscular hyperactivity, such as strabismus, upper limb spasticity, oromandibular and cervical dystonia, etc. [6].

Transient, reversible synaptic silencing has been the basis of BoNT-A clinical efficacy and safe use. In individual patients, the treatment duration (the time period between the toxin’s application and the cessation of its beneficial actions requiring reinjection) may depend on several factors related to appropriate targeting, dosing and mechanism of the toxin action, as well as the mechanism of the underlying disease [7]. Ideally, an appropriate mode of toxin application and dosage should produce sufficiently lasting beneficial actions with minimal local or systemic adverse effects [8]. In the muscles targeted for cosmetic or neurological use, the primary toxin action involves an initial prominent blockade of motor unit activity, followed by remodeling of the neuromuscular junction and reversible recovery of the quantal acetylcholine (ACh) release at the original muscular endplate zone [9]. Careful muscle targeting with appropriate techniques (e.g., muscle identification by electrophysiology and ultrasound and endplate targeting by appropriate toxin dilution and volumes) provides optimal and reliable clinical effects emphasizing the importance of peripheral muscular action [10,11]. However, the toxin treatment may exert a variable extent and duration of antispastic or anti-dystonic action in comparison to separately assessed local neuromuscular paralytic actions. Several clinical reports suggest that the toxin’s beneficial actions may surpass the duration of the peripheral muscle weakness. On the other hand, patients may experience an unsatisfactory therapeutic duration and require sooner reinjections, even if local muscular action is still present [12,13,14]. Clinical evidence suggests that BoNT-A modifies muscle spindle activation and the consequent proprioceptive afferent input associated with the exaggerated stretch reflex, either directly by the blockage of encapsulated intrafusal terminals or indirectly due to extrafusal muscle relaxation [15,16]. Additional, nonlocal actions have also been hypothesized based on reported BoNT-A neurophysiological effects on distant muscles located far from treated NMJ or the muscle spindle zone. A reduction of spinal recurrent inhibition and the reciprocal facilitation in noninjected muscles of the same limb [17,18,19] and F-wave alterations in hand muscles after cervical muscle treatment [20,21] indicated that BoNT-A may induce distant effect on motor pools, innervating different, noninjected muscles.

Possible direct central effect of the toxin on the neuromotor control has been supported by animal studies. After high-dose BoNT-A injected into ocular or hind limb muscles, a reduced firing activity accompanied by structural alterations of synaptic contacts suggested the toxin’s action on the presynaptic input that controls the motoneuron activity [22,23,24]. The occurrence of BoNT-A-truncated SNAP-25 fragments in central motor and sensory nuclei supports direct central enzymatic actions after toxin axonal transport from the injected peripheral site [25,26,27,28,29]. Moreover, peripherally injected BoNT-A induces proteolytic SNAP-25 cleavage in cholinergic terminals presynaptic to motoneurons [28] and reduced the tetanus neurotoxin (TeNT)-evoked local muscular spasm dependently on transcytosis [30]. However, in the later study [30], central antispastic effects were evident when the toxin was injected into the sciatic nerve, and the translation of this finding to clinically employed intramuscular injections, with strong muscular paralysis possibly masking the central toxin effect, is unclear. Herein, we examined the possibility that the central antispastic action might be influencing the duration of the BoNT-A beneficial antispastic effect once the local muscular toxin action starts to fade. Thus, in rats with a tetanus toxin (TeNT)-evoked transient local muscular spasm, we investigated the toxin’s antispastic activity during the later time period when flaccid paralysis recovered and, further on, the possible involvement of the central action of BoNT-A in the late beneficial antispastic effect.

## 2. Results

### 2.1. Early Effects of BoNT-A on Spastic Paralysis Are Concomitant with Local Muscular Neuroparalytic Effect

Rats developed spastic paralysis starting at 3 days post-TeNT i.m. injection, which peaked at 7 d post-toxin treatment and subsequently started to wear off after day 14 and fully recovered by day 28. The spastic paralysis and lower limb plantar flexor muscle rigidity was evident as a spasm of the injected limb, resulting in limb extension or a reduced range of motion of the tibiotarsal joint. This was evident as the elevated force required for passive ankle dorsiflexion (Figure 1A) and reduced Basso Beattie Bresnahan (BBB) locomotor rating score (Figure 1B). BoNT-A i.m. injections at day 7 post-TeNT fully restored the elevated resistance to passive ankle dorsiflexion already within 1 day after BoNT-A injection, with the effects of different doses of BoNT-A being similar throughout the duration period of TeNT-evoked spastic hypertonia. In addition, all BoNT-A doses similarly restored the locomotor deficit assessed by the BBB scale. 

The muscle-weakening effect of BoNT-A resulting in impaired toe spreading reflex function was quantified, as an increase in the digit abduction score (DAS) was evident already 1 day post-BoNT-A application and started to recover dose-dependently by day 49 post-BoNT-A application. Compared to 5 U/kg, lower BoNT-A doses (2 and 1 U/kg) induced a less intensive and shorter-lasting toe spreading reflex impairment (Figure 2A). In addition to an impaired toe spreading reflex, during the plantar ground placement, we observed the arch-like appearance of the foot and heel weight bearing during the stance in all BoNT-A-treated animals (not shown). The mentioned hind paw appearance change was fully (2 and 1 U/kg) or partially (5 U/kg) recovered by day 49 post-BoNT-A. Weight bearing with plantar surface during initial contact and mid stance recovered by day 49 in all animals, while normal propulsion by using toes and interdigital plantar pads during a terminal stance was only partially recovered in 5 U/kg-treated animals, suggesting some residual local muscular toxin effect from the use of plantar flexion muscles. The atrophic loss of the hind-limb muscle volume was evident as a reduction of the lower leg width at the mid-calf point. Different BoNT-A doses evoked a similar reduction in the lower leg cross-sectional area, which did not show signs of recovery throughout the experiment duration (Figure 2B).

### 2.2. Late Effects of BoNT-A on Spastic Paralysis Are Dose-Dependent and Augmented by Central Toxin Action

The animals that previously recovered from signs of spastic and flaccid paralysis were reinjected with TeNT on day 49 post-BoNT-A intramuscular injection (day 56 post-first TeNT) and redeveloped the focal spasm starting from day 3 after TeNT injection. The antispastic activity of BoNT-A persisted for more than 2 months in a dose-dependent manner. While the antispastic activity of the 5 and 2 U/kg doses reversed the dorsiflexion resistance to the level of the saline-treated vehicle and normalized the locomotor deficit assessed by the BBB scale, the effect of 1 U/kg was significantly lower (Figure 3A), and no significant locomotor improvement was observed (Figure 3B).

In a separate experiment, we assessed the possibility that the late BoNT-A antispastic action is centrally mediated by examining its effect in combination with intrathecally injected BoNT-A-neutralizing antitoxin, which prevents the toxin transcytosis [28,30]. The antispastic action of 5 U/kg BoNT-A on both the dorsiflexion resistance (Figure 4A), as well as the locomotor performance (Figure 4B), was partially reduced by the antitoxin, suggesting the involvement of a central, transcytosis-dependent toxin action. The atrophic reduction of the hind-limb diameter and estimated cross-sectional area was not significantly affected by the intrathecal antitoxin treatment, suggestive of a lack of involvement of a central action of the toxin on muscle atrophy (Figure 4C).

### 2.3. Lasting Effects of BoNT-A Are Accompanied by Toxin’s Ongoing Enzymatic Activity in Both, Injected Muscle and Spinal Cord

After observing that the behaviorally assessed beneficial antispastic actions of BoNT-A persist for more than 2 months after a single treatment in animals, we assessed the possible persistence of BoNT-A–cleaved SNAP-25 (cSNAP-25) immunoreactivity in injected gastrocnemius muscles and the spinal cord. The enzymatic activity in NMJ persisted up to day 71 post-BoNT-A injection at all BoNT-A doses injected (Figure 5A). In line with the nonuniform distribution of NMJs in the examined muscle, the staining of cSNAP-25 was not visible in all visual fields but within the single longitudinal section of gastrocnemius, it was easy to identify elongated motor endplate zones containing cSNAP-25 immunoreactivity at all the examined doses. A semiquantitative evaluation showed a similar occurrence of NMJs and nerve terminals in the examined visual fields that contained the staining, with a slightly lower number of stained NMJs and axons in 1 U/kg-treated muscle (Figure 5C).

In addition to long-term synaptic silencing in the periphery, the product of BoNT-A enzymatic activity was also present in the spinal cord ventral horn (Figure 5B). The area of cleaved SNAP-25 immunoreactivity in the muscle in the ipsilateral and contralateral spinal cord ventral horn was shown to be dose-dependent (Figure 5D). The cleaved SNAP-25 surrounding the ventral horn motoneurons did not colocalize with MAP-2 immunoreactivity, suggesting the lack of toxin activity in the somatodendritic compartment belonging dominantly to motoneurons (Figure 5E).

## 3. Discussion

To account for the BoNT-A therapeutic action in hyperactive muscle, it is widely assumed that the desirable therapeutic benefit is entirely due to the toxin’s well-known long-term neuroparalytic activity at peripheral muscular cholinergic terminals as its natural, dominant target. Hence, possible actions at additional levels of the motor nervous system, including CNS, might have been supposed to be unwarranted and unnecessary. However, clinical observations suggesting that the period of neuromuscular paralysis does not necessarily coincide with the treatment duration, or that the patients may benefit from doses producing little or no muscle-weakening effect, and alterations of neurophysiological activity of noninjected muscles exceed the simple functional role of neuromuscular paralysis [12,13,14,31,32]. Herein, we found that the maximal beneficial actions in rat spastic muscle may persist beyond the local neuroparalytic effect with participation of the central toxin action.

A preclinical assessment of the muscle-weakening effect, i.e., toxin-mediated flaccid paralysis in the normal muscle of non-spastic animals or isolated ex vivo preparations, is considered a correlate of its antispastic action, thus, overlooking possible additional mechanisms of action relevant only for pathological muscle hyperactivity. In line with that, studies of BoNT-A action in experimental disorders of elevated muscle activity have been scarce [29,33,34]. In the present study, we employed repeated low-dose TeNT i.m. injections to induce local spastic paralysis mimicking the disinhibition of the synaptic input to motor neurons in spasticity and hyperkinetic movement disorders responsive to commonly employed antispasmodic baclofen, as well as BoNT-A [30]. Herein, we examined the long-term effects of intramuscularly injected BoNT-A by behavioral scoring methods for the evaluation of both flaccid and spastic paralysis and locomotor deficits in rodents. The chosen peripheral BoNT-A doses (1–5 U/kg) in rats corresponded to low–moderate doses commonly employed in clinical practice (corresponding to 70–350 U in an average 70 kg human).

During the early period post-BoNT-A (injected 7 d post-TeNT), we observed the fast and complete reversal of TeNT-mediated spastic paralysis and locomotor deficits evident as increased dorsiflexion resistance or an impaired BBB locomotor scale (Figure 1). The early onset of BoNT-A antispastic action cooccurred with the toxin-mediated toe spreading reflex impairment (assessed by DAS), which started within 24 h (Figure 2A). This is in line with the commonly observed fast clinical onset of BoNT-A action starting together with a neuromuscular paralytic effect [8]. In line with clinical observations that higher doses are not necessarily related to a higher therapeutic efficacy of BoNT-A [35], we found no dose-dependent differences in the early BoNT-A antispastic activity (Figure 1). The DAS score impairment was, however, shown to be dose-dependent in terms of the peak intensity and duration (Figure 2A), keeping in mind that this impairment is indicative of BoNT-A diffusion into the noninjected nearby muscles mediating the toe abduction rather than the local action within the injected gastrocnemius [36,37]. The 1–5 U/kg doses employed here are expected to produce substantial to near-complete muscular paralysis of gastrocnemius in the immediate period post-BoNT-A [27,38], which may account for the lack of dose-dependent difference in the BoNT-A antispastic action in the early time period post i.m. treatment. The gastrocnemius CMAP recovers up to 30–35% of the original value by day 35 and up to 68% after 84 days post-i.m. 2.5 U BoNT-A [38]. This suggest that a significant recovery of muscular function (albeit slower and incomplete compared to DAS) is expected after 1 to 2 months at the doses examined here. In line with that, the arch-like appearance of the paralyzed hind paw or weight bearing by employing the heel during the stance was recovered in all BoNT-A-treated animals (not shown). In contrast to the significant functional recovery of the hind-limb appearance and use, the atrophic reduction of the lower leg muscles was similar at all doses employed here, and notable recovery was not observed throughout the experiment (Figure 2B and Figure 4C).

To study the late effect of BoNT-A, we reinstated the muscle hypertonia by using a second TeNT injection on day 49 after the BoNT-A pretreatment. The higher two doses (5 and 2 U/kg) maintained their antispastic action, normalizing the muscle hypertonia and locomotor deficit for more than two months post-BoNT-A. On the other hand, the lowest examined 1 U/kg dose lost its ability to block the TeNT-evoked locomotor deficit assessed by the BBB and was shown to be less effective in reducing the elevated muscle tone (Figure 4A,B). Thus, our results are in line with clinical observations reporting a longer treatment duration with higher doses. A comparative study of the beneficial effects of BoNT-A (abobotulinumtoxinA) on cervical dystonia, assessed by a modified Tsui severity scale, suggested the longest duration of effect and lowest need for reinjection after 8 weeks in the highest dose-treated patient group (1000 U total dose vs. lower dose groups (250 and 500 U)) [39]. Compared to onabotulinumtoxinA, abobotulinumtoxinA administered at a higher conversion ratio (3:1) was more efficacious than the same preparation administered at a 1.7:1 ratio at 12 weeks post-treatment [40]. However, a careful balance between the doses employed, as well as other factors such as adverse local or systemic effects of a given treatment, should be taken into account before considering the use of higher toxin doses in patients.

Based on the findings that BoNT-A-mediated therapeutic benefits are not concurrent or outlast the flaccid paralysis [31,32,41], we hypothesized that a central action may contribute to the overall toxin-mediated benefit when the muscular effects start to fade. Thus, after observing that a BoNT-A-mediated antispastic effect might persist for 2 months post-BoNT-A despite the functional neuromuscular recovery, we further went on to examine the possible contribution of the central toxin action. In additional animals treated with 5 U/kg BoNT-A, we blocked the toxin’s transcytosis by a BoNT-A-neutralizing antibody administered intrathecally 24 h post-BoNT-A (and not at the same time to avoid any possible interference with BoNT-A binding and entrance at the neuromuscular endplate zones) and then injected the same muscle with TeNT at day 50 post-BoNT-A to establish muscle hypertonia. Importantly, to be able to interpret the results of the antitoxin experiment, it has to be pointed out that the animals treated with i.m. BoNT-A (and i.t. horse serum as a control for i.t. equine antitoxin) exerted unopposed BoNT-A activity at both the peripheral muscular and central spinal sites, while the BoNT-A + antitoxin- treated animals should have an unopposed peripheral toxin effect and central effect, depending only on the transcytosis being prevented by the neutralizing antitoxin. We found that the antitoxin partially counteracted the beneficial effects of BoNT-A on the TeNT-evoked elevated muscle tone and locomotor deficit, which suggests a combination of peripheral and central toxin actions simultaneously contributing to the observed late antispastic activity of BoNT-A. The possibility that the direct central toxin effects contribute to the observed overall beneficial effect adds to our understanding of the mechanisms of this toxin’s long-term central actions.

At the end of the behavioral assessment, we examined the persistence of the toxin’s enzymatic activity in both the injected muscle and the corresponding spinal cord segment. In line with the short turnover half-time of synaptic proteins such as SNAP-25 lasting only few days, this points to the ongoing presence of SNAP-25 cleaving protease for over 2 months post-toxin i.m. injection. Thus, despite the obvious functional muscle recovery, our results are in line with the long-term muscular actions of enzymatically active BoNT-A. In the spinal cord ventral horn, central SNAP-25 cleavage indicated the continuous central presence of BoNT-A protease as well. At the dose range employed here, the quantity of cleaved SNAP-25 in the CNS was shown to be dependent on the amount of injected BoNT-A. Although all BoNT-A doses employed here produced similar antispastic actions during the early period of BoNT-A action, only the higher two doses (5 and 2 U/kg) continued to show the full antispastic effect at a later time period. Theoretically, as the level of active BoNT-A protease at the relevant synaptic sites in both the muscle and spinal cord drops with time, a higher BoNT-A dose may maintain its antispastic effect for a longer time period. In other central motor regions, its lasting action on the basal ganglia circuitry and reduction of the motor deficits in experimental parkinsonism has been demonstrated after its direct injection into the striatum [42,43], which is most likely associated with the ongoing BoNT-A enzymatic activity lasting for several months after the toxin central injection [25].

In the ventral horn, MAP-2 immunostaining of the somatodendritic compartment dominantly belonging to the motoneurons did not colocalize with BoNT-A-cleaved SNAP-25, in line with the evidence that BoNT-A holotoxin physically leaves the motor neurons and enters the presynaptic axonal compartment of the second-order neurons [28,29]. In the ventral horn and brainstem motor regions, cSNAP-25’s distinctive localization within large synaptic terminals containing choline-acetyltransferase and vesicular acetylcholine transporter often in contact with motoneuronal cell bodies suggested toxin action within the cholinergic C-boutons [26,28,44], a premotor synaptic input from V0_C_ lineage-derived excitatory interneurons that support sustained motoneuronal firing during intense motor activities (e.g., swimming) [45]. This is in line with the possibility that BoNT-A reduces the motoneuronal excitability by inhibiting the excitatory cholinergic drive onto motoneurons, which, apart from immunohistochemical evidence, has not been conclusively demonstrated yet. The involvement of additional synaptic sites expressing lower levels of cleaved SNAP-25 and/or the simultaneous synaptic blockade of other central neurotransmitters cannot be ruled out as well.

## 4. Materials and Methods

### 4.1. Animals

Male Wistar Han rats (Department of Pharmacology, University of Zagreb School of Medicine), 2.5–3 months old and weighing 350–400 g at the beginning of experiment, were used in all the experiments. The animals were kept under a 12-h light and dark cycle regime with unlimited access to the standardized diet (Mucedola Srl., Milano, Italy) and drinking water. The experiments were approved by the review boards of the University of Zagreb School of Medicine and Croatian Ministry of Agriculture (No. EP 229/2019).

### 4.2. Pharmacological Treatment

For intramuscular neurotoxin injections, the rats were briefly anesthetized with isoflurane (Forane, Abbot Laboratories, Abbot Park, IL, USA), 5% induction and 1–1.5% maintenance by using a gas anesthesia system for small animals (Ugo Basile, Varese, Italy). The rats were intramuscularly injected with 1.5 ng tetanus neurotoxin (TeNT, Sigma, St. Louis, MO, USA) and BoNT-A (INN: Clostridium botulinum type A neurotoxin complex, Allergan Inc., Irvine, CA, USA). Different doses of BoNT-A were employed: 1, 2 or 5 international units per kg (U/kg, each unit consisting of 48 pg of a 900-kDa toxin complex). The neurotoxins were injected into the right gastrocnemius in a total volume of 20 µL divided into the medial and lateral muscle belly (10 µL each) by employing a 50-µL Hamilton syringe as previously described [30]. To block the central BoNT-A transcytosis, animals were injected with BoNT-A-neutralizing antitoxin (National Institute for Biological Standards and Control, NIBSC code 14/174, Potters Bar, UK; a kind gift from Dr. Thea Sesardic, 10 I.U. in 10-μL volume) into the intrathecal lumbar space at the level of cauda equina by employing a 0.3-mL 27 G tuberculin syringe (BD, Franklin Lakes, NJ, USA). The rats were anesthetized by intraperitoneal (i.p.) ketamine and xylazine (70 and 7 mg/kg, respectively). Then, the rat’s hair covering the lumbar spine was clipped, and a small skin incision was made to visualize the midline joining area of the paraspinal muscles. Subsequently, a needle was advanced between the lumbar vertebrae into the spinal canal, which was confirmed by brief tail or hind-leg movement as the needle touched the spinal nerve roots within the cauda equina. Next, the antiserum or horse serum was injected slowly into the lumbar space and the needle carefully withdrawn after 10–15 s. The incision was joined by 1 to 2 sutures using the 5–0 surgical thread (Mersilk, Ethicon, Cincinnati, OH, USA) and disinfected by povidone-iodine (Alkaloid, Skopje, North Macedonia).

### 4.3. Behavioral Motor Assessment

#### 4.3.1. Digit Abduction Score

To measure the local flaccid paralysis of the hind-limb, the reflex toe abduction was used to evaluate the digit abduction score (DAS), similar to as previously described [36]. The rats were lifted by their waist, and then, the DAS score was determined as the number of hind-paw toes that could not abduct normally after the loss of contact with the ground (0 = all fingers abduct from each other, 1 = 2 toes cannot abduct, 2 = 3 toes cannot abduct, 3 = 4 toes cannot abduct and 4 = no toe abduction).

#### 4.3.2. Resistance to Ankle Dorsiflexion

We measured the intensity of the TeNT-evoked muscular rigidity by assessing the force needed to overcome the resistance of spastic hind-paw plantar flexors during passive ankle dorsiflexion in trained calm animals, as previously described [29]. In brief, the animals were held around the waist with the experimenter’s hand gently pressing the interdigital paw pad against an elevated plastic platform mounted on a digital kitchen scale to flex the spastic joint. When the 90° tibiotarsal angle was achieved or slightly exceeded, the pressure exerted by the experimenter was slightly relieved, and the resistance in grams was noted just before the point when further pressure relief would return the tibiotarsal angle to values above 90°. The average of two values per limb was taken per each single measurement.

#### 4.3.3. Basso Beattie Bresnahan Locomotor Scale

To assess the hind-limb usage and the range of motion of hind-limb joints during walking, we employed a modified version of the Basso Beattie Bresnahan (BBB) locomotor score [46] originally devised to assess the motor effect of experimental spinal cord injuries. The BBB scale consists of 21 defined grade points assessing the range of motion of hind-limb joints, stepping, coordination, paw position, trunk stability and tail position in their gait. Presently, the non-spastic left limb contralateral to i.m. TeNT was not included into the quantification to obtain the overall score; however, it conveniently served as a reference for normal non-spastic joint range of motion during gait. The rats were allowed to walk across a flat black table surface and freely return to their home cage with its opened top placed level with the table. The videos were taken with a web camera with an unopposed view of the spastic limb during walk across the table towards the cage. The BBB score was assessed offline by two independent observers unaware of the animal treatment, and the two scoring results were averaged as a single animal measurement.

#### 4.3.4. Measurement of Lower Leg Muscle Atrophy

Gastrocnemius injection of BoNT-A results in notable muscle reduction and visible thinning of the lower leg. Thus, throughout the course of the experiment, we estimated the muscle atrophy by noninvasive external measurement of the hind-leg diameter by using a digital caliper. The mediolateral diameter value was noted at the widest point of the rat calf belly with the jaws of the caliper gently touching the skin of both sides perpendicularly to the Achilles tendon while dorsoventral diameter was noted at the same level by touching the tibial bone on one side and the skin overlaying the middle of the calf on the other side. The approximate cross-sectional area of the rat lower leg was calculated by using the formula for the area of an ellipse (in mm^2^) defined by the two diameters obtained (dorsoventral × mediolateral × π/4).

### 4.4. Time-Course of Experiments

#### 4.4.1. Experiment No 1. The Dose-Response Assessment of Early and Late BoNT-A Antispastic Action

The animals were assigned to 5 experimental groups (N = 8/group) and treated i.m. with TeNT or the 2% BSA-containing saline vehicle. After the development of spasticity, on day 7 post-TeNT, the animals were treated with different BoNT-A doses (1, 2 and 5 U/kg) or physiological saline (control non-spastic and spastic groups). The motor behavioral effect of the toxins was assessed at different time points (pretreatment, days 3 and 7, and then after BoNT-A i.m. days 8 and 10 post-TeNT (on days 1 and 3 post-BoNT-A). From then on, the measurements were conducted once a week until day 56 post-TeNT/49 post-BoNT-A, when another 1.5 ng TeNT injection was made to reinduce the spasticity. Then, afterwards, the measurements were repeated on days 3, 7, 12 and 19 post-the second TeNT (days 52, 56, 61 and 68 post-BoNT-A)

#### 4.4.2. Experiment No 2. Assessment of the Role of Central Action and Transcytosis in the Late BoNT-A Antispastic Activity

The rats (N = 6/group) were injected with 5 U/kg BoNT-A and then, after 24 h, with BoNT-A-neutralizing antitoxin into the lumbar intrathecal space, as described in the previous section. On day 50 post-BoNT-A, the animals were injected with TeNT into the right gastrocnemius. Then, the measurements were repeated on days 3, 7, 12 and 19 post-the second TeNT injection (days 53, 57, 62 and 69 post-BoNT-A).

### 4.5. Immunohistochemistry

The animals from experiment no. 1 were deeply anesthetized with ketamine-xylazine and sacrificed by transcardial perfusion with physiological saline, followed by cold 4% paraformaldehyde + phosphate-buffered saline (PBS) fixative. The gastrocnemius muscle and spinal cord were excised, post-fixed and cryoprotected overnight in 15% sucrose + fixative and, the next day, transferred to 30% sucrose in 1 × PBS. Then, after the tissue sank, it was removed from sucrose + PBS and kept in an ultra-freezer at −80 °C. The muscles were cut in the cryostat at a 20-μm per slice thickness and immediately transferred to glass adhesion slides (Super Frost Plus Gold, Thermo Scientific, Waltham, MA, USA), while the spinal cord slices were cut at 35-μm slices and placed in free-floating wells.

The muscles were washed with PBS + 0.25% triton-X-100 (PBST), blocked in 10% NGS and incubated overnight at room temperature with nonaffinity purified rabbit polyclonal antibody to BoNT-A-cleaved SNAP-25 recognizing the SNAP-25 1–197 fragment (1:4000, National Institute for Biological Standards and Control, Potters Bar, UK), followed the next day by 1:400–500 goat anti-rabbit Alexa 555 secondary antibody (Cell Signaling, Danvers, MA, USA). Primary and secondary antibodies were diluted in 1% NGS and PBST. After that, the slides were coverslipped with antifading agent. The cleaved SNAP-25 occurrence in muscles was quantified by a previously reported semiquantitative method [37] assigning a defined score to observable neuromuscular junctions, nerve terminals and nerve profiles. Score 1 was assigned when only a few or weak NMJ staining was observed; score 2 = the moderate staining of frequent NMJs; score 3 = the strong staining of frequent NMJ and nerve terminals and score 4 = the strong staining of frequent NMJs, terminals and intramuscular nerves (no. of axons ≥ 10). The individual score assigned for each animal was derived from the average score of 4 visual fields, each taken from different muscle slices at 20 × 0.5 magnification. Spinal cord slices were incubated with 3% H_2_O_2_ for the inhibition of endogenous peroxidase and washed in PBS. Then, the slices were blocked with 10% NGS and incubated overnight at 4 °C with mouse monoclonal antibody to somatodendritic marker microtubule-associated protein 2 (MAP-2) (Sigma Aldrich, St. Louis, MO, USA, 1:1000 dilution in 1 % NGS). The next day, the tissue was incubated with goat anti-rabbit Alexa 555 secondary antibody (Cell Signaling, Danvers, MA, USA) and washed. After that, the spinal cord sections were incubated with the antibody to BoNT-A-cleaved SNAP-25 (1:8000) overnight at room temperature. The following day, the tissue was incubated with poly-HRP-conjugated goat anti-rabbit secondary antibody and then reacted with tyramide Atto-488 HRP substrate prepared as described previously [47] diluted 1:100 in PBS with 100 mM imidazole and 0.001% H_2_O_2_ for 10 min. After that, the slices were washed 3 times, mounted on glass slides and coverslipped with antifading agent. Then, the images were visualized by an Olympus BX-51 fluorescent microscope and processed using Cell Sens Dimension imaging and quantification software (Olympus, Tokyo, Japan). The cSNAP-25 immunoreactivity was quantified as reported previously [30]. If brief, microphotographs taken at 40 × 0.5 magnification were acquired at constant exposure in the ipsilateral and contralateral ventral horns. Then, the area containing the cSNAP-25 immunoreactivity was quantified from separated green channel images by using a constant pixel threshold range (100–256 by employing cell Sens Dimension software). The value representing the ventral horn cSNAP-25 immunoreactive area was calculated as the total area from 3 nonoverlapping visual fields (3 × 0.14 mm^2^) and the average of 4 slices per single animal. Samples with MAP-2/cSNAP-25 colocalization were visualized with a Leica SP5 Confocal microscope equipped with a 40× HCX PL APO NA 1.4 oil immersion and, afterwards, processed by ImageJ without altering the intensity of the signals.

### 4.6. Statistical Analysis

The data were presented as mean ± SEM or individual values with the median. Repeated measurements were analyzed by two-way analysis of variance (two-way RM ANOVA), followed by Bonferroni’s post hoc test for between-group comparisons. The statistical analysis of the results of the cleaved SNAP-25 immunoreactive area was not normally distributed; hence, nonparametric ANOVA and Dunn’s post hoc were employed. A *p*-value lower than 0.05 was considered significant.

## 5. Conclusions

In rat model of local spasticity, we found that the persistence of BoNT-A effect and the relative importance of central toxin action change depending on the doses injected and the time point examined. In line with clinical experience, local muscular paralysis is the dominant factor contributing to the initial antispastic effect of i.m. BoNT-A, explaining its early onset and intensity. Later, when peripheral muscular effects start to fade, central antispastic actions become a more apparent contributor to the overall BoNT-A-mediated benefit. These results suggest the relevance of peripheral doses employed and central toxin action as additional factors contributing to extraordinarily long BoNT-A clinical action.

## Figures and Tables

**Figure 1 ijms-23-11626-f001:**
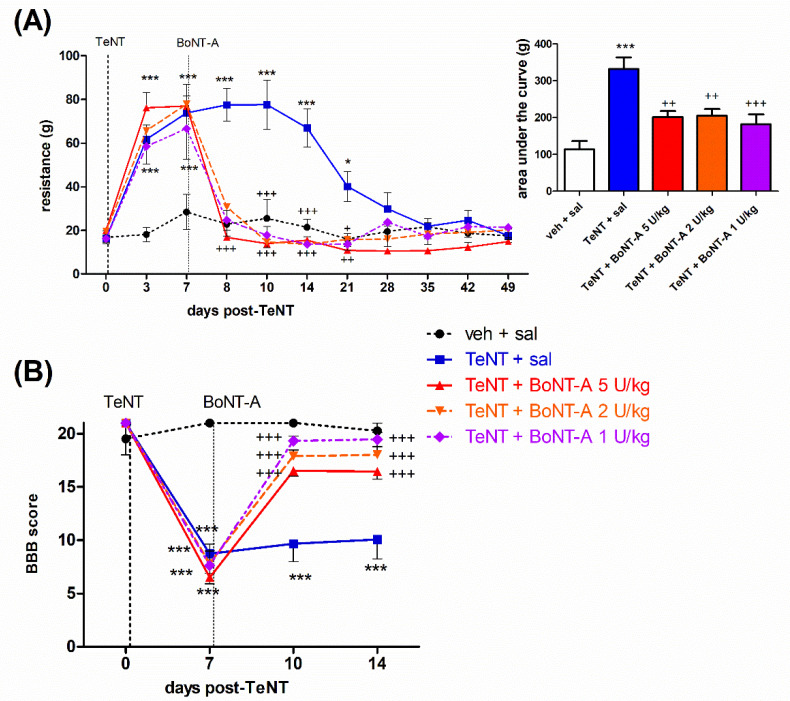
The fast-onset antispastic effect of botulinum toxin type A (BoNT-A) on tetanus toxin (TeNT)-evoked hind limb rigidity and locomotor deficit in rats. The BoNT-A (1–5 U/kg) intramuscular (i.m.) injection into gastrocnemius reduces the preestablished TeNT (1.5 ng, i.m.)-evoked resistance to ankle dorsiflexion (reaching a 90° tibiotarsal angle) (**A**), and the locomotor deficit assessed by the Basso Beattie Bresnahan (BBB) scale (**B**). Veh, vehicle; sal, saline i.m. treatment. Horizontal bars indicate the time points of TeNT and BoNT-A i.m. application. N = 8 to 9 animals/group; mean ± SEM, * and ***: *p* < 0.05 and < 0.001 vs. veh + sal, +, ++ and +++: *p* < 0.05, < 0.01 and < 0.001 vs. TeNT + sal (two-way RM ANOVA, followed by Bonferroni’s post hoc test; *p* < 0.05 considered significant).

**Figure 2 ijms-23-11626-f002:**
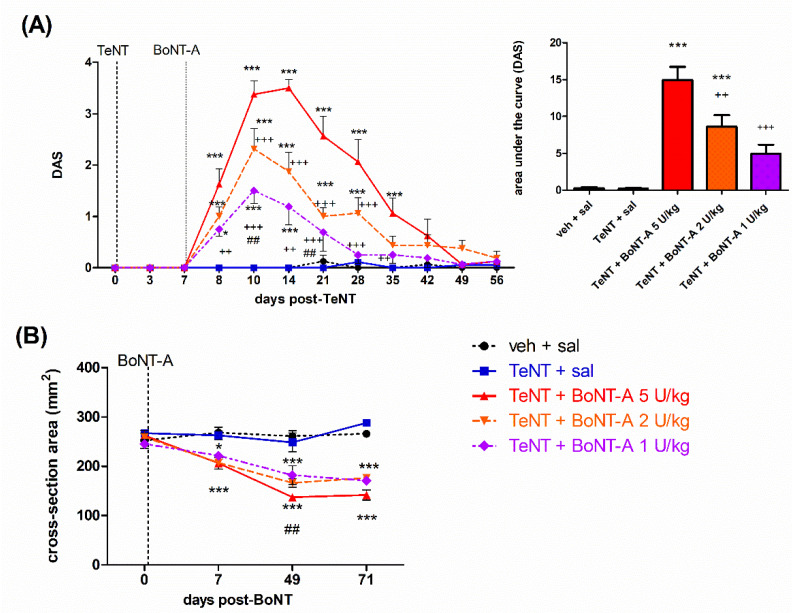
Intramuscular BoNT-A induces reversible impairment of local hind-limb muscle function and non-recovering atrophy in rats. The BoNT-A (1–5 U/kg) intramuscular (i.m.) injections into gastrocnemius induces dose-dependent lasting toe spreading reflex inhibition (**A**), assessed by the digit abduction score (DAS), and atrophic reduction of the lower leg muscles (**B**), represented as the estimated cross-section area of the lower leg at the middle calf muscle level (calculated as the area of ellipse defined by dorsoventral and mediolateral leg diameters). Veh, vehicle; sal, saline i.m. treatment. Horizontal bars indicate the time points of TeNT and BoNT-A i.m. application. N = 8 to 9 animals/group; mean ± SEM, * and ***: *p* < 0.05 and < 0.001 vs. veh + sal, ++ and +++: *p* < 0.01 and < 0.001 vs. TeNT + sal; ##: *p* < 0.01 vs. TeNT + BoNT-A 1 U/kg (two-way RM ANOVA, followed by Bonferroni’s post hoc test; *p* < 0.05 considered significant).

**Figure 3 ijms-23-11626-f003:**
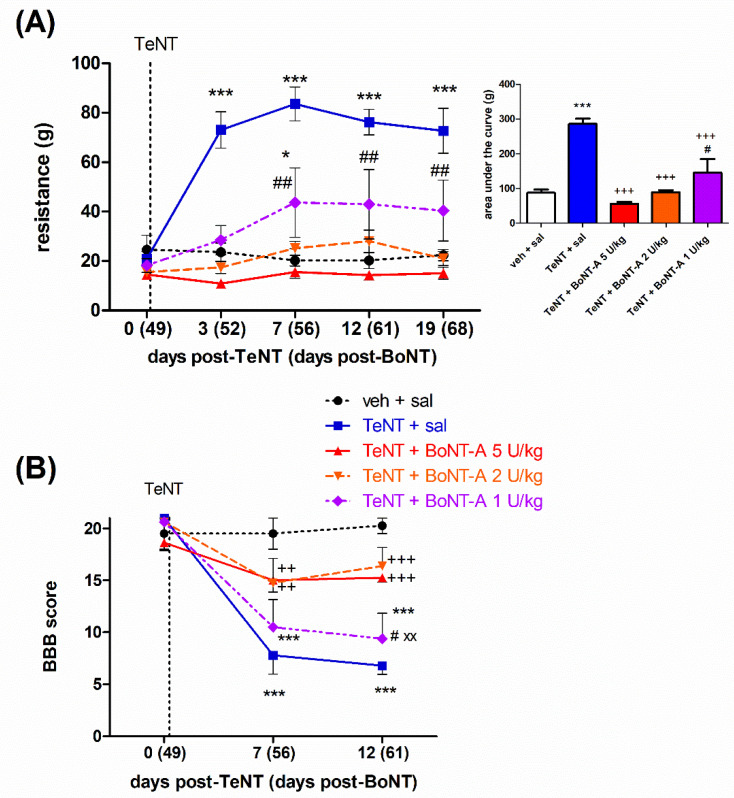
The late dose-dependent BoNT-A antispastic effect persists two months post-i.m. treatment. The effect of BoNT-A (1–5 U/kg) intramuscular (i.m.) injection into the gastrocnemius on a TeNT (1.5 ng, i.m.)-evoked local muscle spasm reinduced 49 days post-BoNT-A was assessed by measuring the resistance to passive ankle dorsiflexion reaching a 90° tibiotarsal angle (**A**) and the locomotor deficit assessed by BBB scale (**B**). Veh, vehicle; sal, saline i.m. treatment. Horizontal bars indicate the time point of TeNT i.m. application. N = 8 to 9 animals/group; mean ± SEM, * and ***: *p* < 0.05 and < 0.001 vs. veh + sal; ++ and +++: *p* < 0.01 and < 0.001 vs. TeNT + sal; # and ##: *p* < 0.05 and < 0.01 vs. TeNT + 5 U/kg BoNT-A; XX: *p* < 0.01 vs. TeNT + 2 U/kg BoNT-A (two-way RM ANOVA, followed by Bonferroni’s post hoc test; *p* < 0.05 considered significant).

**Figure 4 ijms-23-11626-f004:**
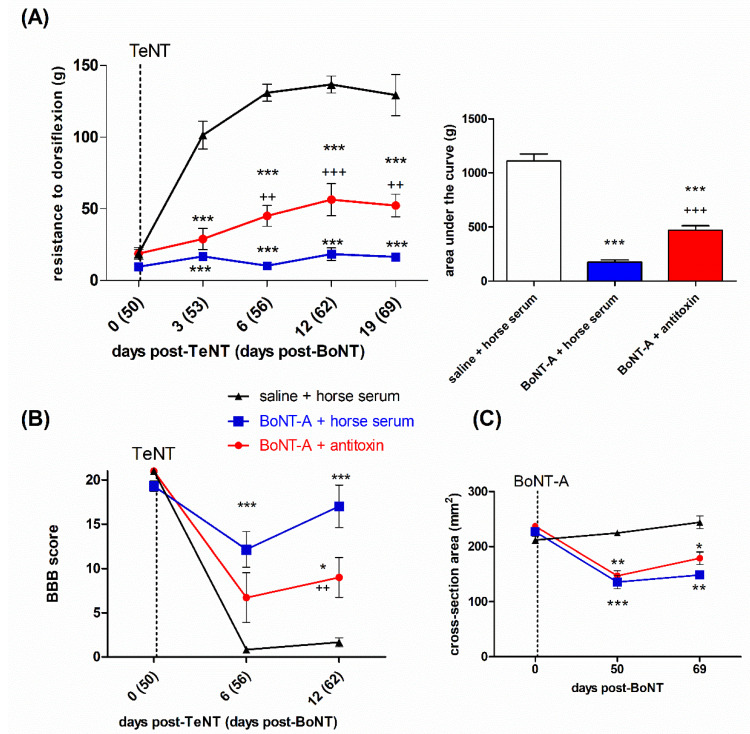
The late BoNT-A antispastic effect involves transcytosis-dependent central action. The rats were treated with i.m. BoNT-A (5 U/kg) and, then, after 24 h, injected i.t. into the lumbar spinal canal with the neutralizing equine antitoxin (or horse serum as the control) to prevent its central transcytosis-dependent action. Then, after a further 49 days, its effect on the TeNT (1.5 ng, i.m.)-evoked local muscle spasm was assessed by measuring the resistance to ankle dorsiflexion (reaching a 90° tibiotarsal angle) (**A**) and the locomotor deficit assessed by the BBB scale (**B**). The central toxin action does not affect the atrophic loss of the lower leg estimated cross-sectional area at the widest calf point (**C**). Sal, saline i.m. treatment. Horizontal bars indicate the time points of TeNT i.m. application. N = 6 to 7 animals/group; mean ± SEM, *, ** and ***: *p* < 0.05, < 0.01 and < 0.001 vs. sal + horse serum, ++ and +++: *p* < 0.01 and < 0.001 vs. BoNT-A + horse serum (two-way RM ANOVA, followed by Bonferroni’s post hoc test; *p* < 0.05 considered significant).

**Figure 5 ijms-23-11626-f005:**
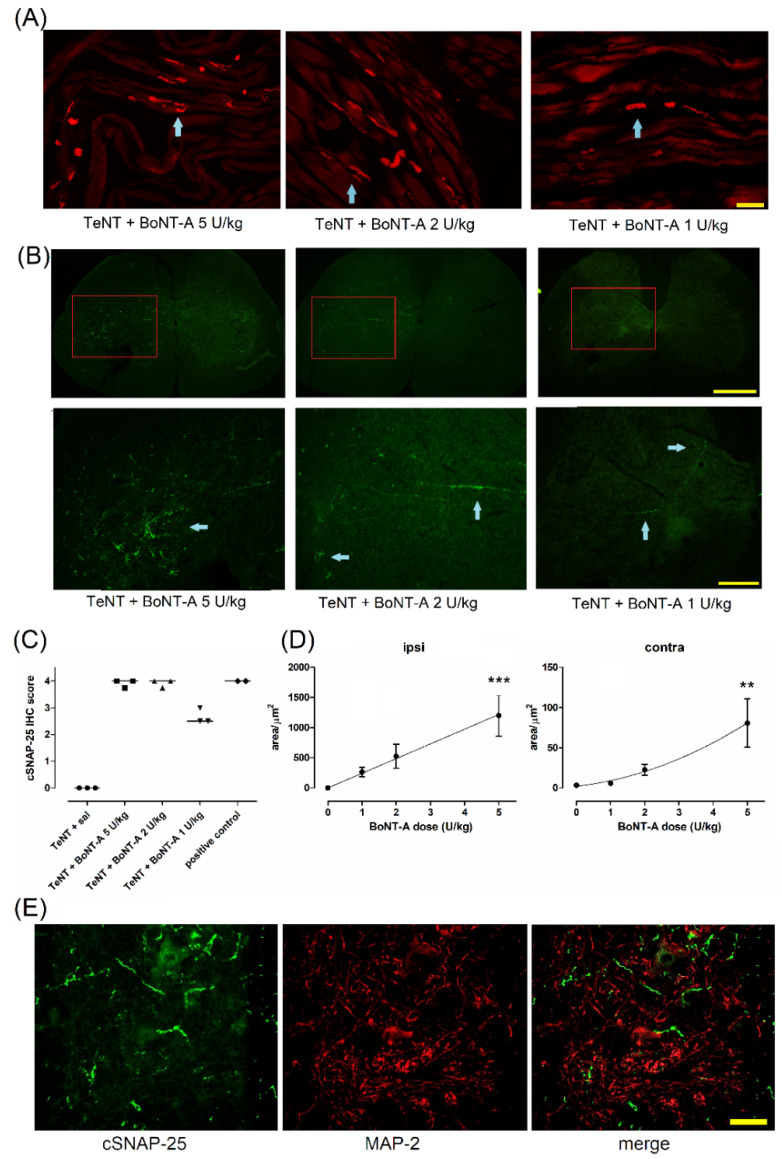
Enzymatic product of BoNT-A in the muscle and spinal cord persists for more than 2 months post-BoNT-A i.m. treatments. BoNT-A (1–5 U/kg) was injected into the gastrocnemius muscle, and the presence of BoNT-A-cleaved SNAP-25 (cSNAP-25) was examined in the injected muscle (**A**) and spinal cord (**B**) on day 71 post-toxin treatment. (**A**) Red immunoreactivity shows neuromuscular junctions and nerve terminals (cyan arrow; scale bar = 100 µm). (**B**) Upper and lower-panel microphotographs indicate the low (2×) and higher (5×) magnification images of the same representative L4 spinal cord sections with green immunoreactivity indicating cSNAP (arrow) (scale bars = 500 µm and 200 µm, respectively). The scoring method suggests an abundant presence of neuromuscular junctions, nerve terminals or axons immunoreactive for cleaved SNAP-25 in BoNT-A-treated muscle at more than 2 months post-BoNT-A (N = 3 animals per group, 4 visual fields per animal each taken from different section), as well as in 2 positive controls (obtained from the muscle treated with 5 U/kg BoNT-A 7 days prior to perfusion (**C**)). Under (**C**), the data points represent individual animal score values and the horizontal bar represents median. The immunoreactive cSNAP-25 area quantity in the ipsilateral (ipsi) and contralateral (contra) ventral horn is dose-dependent (**D**). The data were obtained by analysis of the pixel intensity threshold area of cleaved SNAP-25 immunoreactivity in 3 nonoverlapping high-magnification (20×) visual fields (433 μm × 323 μm = 0.14 mm^2^) located in the lateral L4 ventral horn, average of 4 slices per animal (N = 5 to 6 animals per group; mean ± SEM, ** and ***: *p* < 0.01 and < 0.001 vs. BoNT-A dose = 0 from TeNT + saline animals (Kruskal–Wallis test, followed by Dunn’s post hoc). In the L4 ventral horn, the cSNAP-25 does not colocalize with the somatodentritic compartment immunoreactive for microtubule-associated protein 2 (MAP-2) (**E**). The confocal microscope image was obtained from animals injected with 5 U/kg BoNT-A (scale bar = 50 µm).

## Data Availability

The data are contained in the article.

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
