# Peer review of "Lasting Peripheral and Central Effects of Botulinum Toxin Type A on Experimental Muscle Hypertonia in Rats"

_ijms, 2022, doi:10.3390/ijms231911626_

Round 1

Reviewer 1 Report

In this paper authors report a study examining the persistence of BoNT-A antispastic effects on a rat model of local spasticity, and correlate this action with the possible central action of toxin . The purpose of the research was an attempt to respond the unsolved question about the understanding of which is the mechanism of action of BTX-A at the level of central nervous system, one of the most debated issue in the field of botulinum neurotoxin.  To solve this intriguing question, authors used two experimental approaches: behavior and immunofluorescence assays. The experimental approach seems correct, the methods used appear appropriate, and the results are congruent with the hypothesis formulated by the authors. Particularly innovative is the animal model based on the interaction between the effect of tetanous toxin, used as inducer of tetanic spasms in peripheral system, and the botulinum neurotoxin used as counteractor of the induced tetanic spasticity. As I have not encountered specific problems in this paper, it may be accepted for future publication in Toxins after only minor corrections listed below.

Minor:

1) Please change the thickness of error bars in Fig. 4. In fact if you compare they with error bars in other figures they appear more thickness.

2) Color arrows in fig. 5 appear more close to “white” than “blue” as stated in the legend (in panel A they appear cyan, while in panel B they appear white). Please change appropriately.

3) In the block of citations [25-28] on page 74, the paper “Marinelli et al (2012) PLoS One. 2012; 7 (10): e47977” should be cited as a further demonstration of the retrograde transport of the toxin from the peripheral to central nervous system

Author Response

Q1) Please change the thickness of error bars in Fig. 4. In fact if you compare they with error bars in other figures they appear more thickness.

Response: accepted - we changed the error bar thickness according to other figures 

2) Color arrows in fig. 5 appear more close to “white” than “blue” as stated in the legend (in panel A they appear cyan, while in panel B they appear white). Please change appropriately.

Response: Accepted - the image and the figure legend have been changed accordingly. 

3) In the block of citations [25-28] on page 74, the paper “Marinelli et al (2012) PLoS One. 2012; 7 (10): e47977” should be cited as a further demonstration of the retrograde transport of the toxin from the peripheral to central nervous system

Response: accepted - we added the new reference in the manuscript and changed the text accordingly. 

"Occurrence of BoNT-A-truncated SNAP-25 fragments in central motor and sensory nuclei supports direct central enzymatic actions after toxin axonal transport from injected peripheral site [25–29]. "

Reviewer 2 Report

In the present manuscript, a possible central nervous effect of a BoNT-A injection into a muscle was investigated. In particular, it was investigated whether the spasmolytic effect of an intramuscular BoNT-A injection is only due to a flaccid paralysis of the muscle or whether longer-lasting central nervous effects are also involved in this effect. In the context of this question, three further aspects were addressed at the same time: Is BoNT-A transported into the CNS after peripheral injection? How long does this central nervous effect last? Is there a dose dependence of the central effect of a peripheral BoNT-A injection?

The questions were addressed with an elaborate and very well thought-out experimental design. First, spasticity was induced in the right hind paw of the rats by injecting tetanus toxin into the gastocnemius muscle. The experimental animals were subjected to various motor tests to measure spasticity (caused by tetanus toxin) and flaccid paralysis (caused by BoNT-A). Immunohistochemical staining against the cleavage product of BoNT-A, cleaved SNAP-25, was performed on muscle preparations and on specimen of the spinal cord.

Indeed, a long-lasting central effect was measured. It was proven that BoNT-A has a direct central nervous effect after being transported axonally into the spinal cord. The evidence was provided firstly by injecting an antiserum against BoNT-A into the cerebrospinal fluid of the spinal cord of a cohort of experimental animals after the above-mentioned treatments and thus the previously observed long-lasting spasmolytic effect, which was not due to flaccid paralysis, no longer occurred. On the other hand, the cleavage product of BoNT-A, cleaved SNAP-25, could be detected in the anterior horn of the spinal cord by immunohistochemistry. A dose dependence of the effect was demonstrated.

The present manuscript is a cleverly conducted work, the written presentation of the research question, experimental procedure and the results as well as the discussion was done in detail and very well comprehensible. The topic has great scientific and clinical relevance. On the one hand, BoNT-A is now used in everyday clinical practice for a large number of orthopaedic, neurological, internal and cosmetic problems practically everywhere on the human body, on the other hand, in recent years there has been increasing evidence that BoNT-A is transported axonally and can thus also reach the CNS. Little is known about this central nervous effect.

In conclusion, I can recommend this manuscript for acceptance due to its quality and high scientific and clinical relevance and topicality.

However, the discussion should be somewhat more detailed beforehand. Other research groups, for example, have also already observed an effect of BoNT-A lasting for months after direct injection into the CNS of rats and mice. This was mostly done for the experimental treatment of Parkinson's symptoms. For example: Hawlitschka et al. 2018; 2020; Wree et al. 2011

Author Response

Q However, the discussion should be somewhat more detailed beforehand. Other research groups, for example, have also already observed an effect of BoNT-A lasting for months after direct injection into the CNS of rats and mice. This was mostly done for the experimental treatment of Parkinson's symptoms. For example: Hawlitschka et al. 2018; 2020; Wree et al. 2011

Response: Accepted - we extended the discussion and included new references. 

"In other central motor regions, its lasting action on basal ganglia circuitry and reduction of the motor deficits in experimental parkinsonism has been demonstrated after its direct injections into the striatum [42,43], which is most likely associated with the ongoing BoNT-A enzymatic activity lasting for several months after the toxin central injection [25]."